# Providing ambulatory healthcare for people aged 80 and over: Views and perspectives of physicians and dentists from a qualitative survey

**Angélique Herrler**[1,2]*, **Lisa Valerius**[3], **Anna Greta Barbe**[4], **Vera Vennedey**[2], **Stephanie Stock**[2]

**1** Faculty of Human Sciences and Faculty of Medicine, Graduate School GROW–Gerontological Research on Well-being, University of Cologne, Albertus-Magnus-Platz, Cologne, Germany, **2** Institute for Health Economics and Clinical Epidemiology, University Hospital Cologne, Cologne, Germany, **3** Faculty of Management, Economics and Social Sciences, Graduate School GROW–Gerontological Research on Well-being, University of Cologne, Albertus-Magnus-Platz, Cologne, Germany, **4** Centre of Dental Medicine, Department of Operative Dentistry and Periodontology, University Hospital Cologne, Cologne, Germany

* angelique.herrler@uni-koeln.de

**Data Availability Statement:** All relevant data are within the article and its supporting information files. Supplemental file S1 Appendix contains all data on the survey development and pretest.

## Abstract

### Background

People aged 80 and over frequently face complex chronic conditions and health limitations, including oral health problems, which are primarily addressed by ambulatory (i.e., outpatient) healthcare. This demographic development is expected to affect the provision of care. However, few studies have investigated physicians' and dentists' views across the various medical disciplines in non-institutional settings. This study investigated how healthcare providers perceive caring for very old people, and how they feel healthcare should be designed for this patient group.

### Methods

A qualitative online survey comprising nine open-ended items was conducted among physicians and dentists practicing in the ambulatory healthcare sector in North Rhine-Westphalia, Germany. Apart from child and adolescent healthcare, no medical specialties were excluded. The results were analysed using Kuckartz' approach of structuring qualitative content analysis. A descriptive codebook was developed first. After coding all the material, recurring patterns between the topics were investigated and compared between two groups of participants, physicians and dentists.

### Results

N = 77 cases were included in the analysis, from which n = 21 originated from the field of ambulatory general practice care and internal medicine, n = 19 from specialties (e.g., neurology, urology), and n = 37 from dentistry. Caring for patients aged 80 and over was perceived as challenging because of complex health conditions and treatments such as multimorbidity

Supplemental file S2 Appendix contains the full codebook that was developed and used for data analysis. Supplemental file S3 Appendix contains the data obtained from participants in the qualitative survey. Since the data are of a qualitative nature, the authors provided the material only in its coded form and without sociodemographic information on case-level to ensure anonymity. Information on sociodemographic variables is included in Tables 1–2 in the manuscript.

**Funding:** This study was conducted within the Graduate School GROW – Gerontological Research on Well-Being at the University of Cologne, Germany. GROW is funded by the Ministry of Culture and Research of the State of North-Rhine Westphalia, Germany. The authors received no specific funding for this work. The authors acknowledge support for the Article Processing Charge from the DFG (German Research Foundation, 491454339).

**Competing interests:** The authors have declared that no competing interests exist.

and polypharmacy, and patients' cognitive and physical limitations. In characterizing good care in older age, both groups found individual care important, as well as empathetic interactions and good collaboration in networks of healthcare providers. Inadequate reimbursement and limited resources and time were the most important barriers to providing good care, while one of the most important facilitators was healthcare providers' attitude and motivation.

## Conclusions

Physicians' and dentists' conceptions of good healthcare are in line with the conceptualization of patient-centred care. However, the transfer in everyday care delivery is hampered by the current design of healthcare structures. Healthcare providers feel overstrained by the increasing demands placed on them. Adaptations for improvement should focus on building strong networks of cooperating health professions, especially including dental care, and local social support structures.

## Introduction

Due to demographic changes, the section of the population of people aged 80 and over will grow significantly in Europe [1]. Around the age of 80 years, complex long-term health issues such as multimorbidity and frailty show a strong increase, demanding proactive healthcare [2–5] that is primarily provided in the ambulatory (i.e., outpatient) healthcare sector [6,7]. This also applies to the area of dentistry since old age is associated with a variety of oral health issues and dental treatment needs, such as dry mouth problems, periodontal disease, decreased oral function or denture-related conditions [8], while the role of oral health for maintaining the quality of life and well-being is increasingly recognized [9].

However, most healthcare systems are still criticized for being primarily oriented toward acute care that focuses on single conditions and care episodes [10,11]. Therefore, until now, responsibility for the different approaches to healthcare required by the older population has appeared to fall mainly on the micro level and on healthcare providers. This particularly affects providers in the ambulatory healthcare sector, since older people's health matters are frequently of chronic nature, and they have a strong preference for ageing in place and avoiding institutionalisation [12,13]. Moreover, strengthening ambulatory healthcare is one of the main suggestions by institutions such as the Organisation for Economic Co-operation and Development (OECD) or the World Health organisation (WHO) to meet the demands of ageing populations [14,15]. But, apart from geriatric specialisations, approaches to treatment and healthcare delivery for older patients are not yet broadly discussed in medical or dental training, possibly leading to inadequate awareness of the special needs of these patients, and to feelings of stress and frustration in healthcare providers [16,17]. Studies suggest that how professionals perceive older patients and their care has an impact on their patient interactions and satisfaction with their work. For example, it was found that nurses holding negative attitudes towards older people were stereotyping their patients, finding them incapable of decision-making and perceiving them as a burden [18,19]. This conflicts with older people's core motives in healthcare, these being 1) to feel safe, 2) to feel like a meaningful human being, and 3) to maintain control and independence [20]. Therefore, understanding healthcare providers' perceptions and their interactions is crucial for delivering appropriate healthcare to older

adults. However, studies exploring the perspectives and needs of physicians and dentists, apart from a few investigating general practitioners (GPs) [17,21], are rare. What challenges arise in the practice of caring for older adults and what support needs occur, has been primarily examined in institutionalized settings (e.g., [22–24]) or among nurses (e.g., [25,26]). Since older people are especially present in the field of ambulatory healthcare settings with their usual health matters, this poses a considerable gap in understanding the daily reality of patients and healthcare providers, and regarding what is needed to maintain or further develop a proper healthcare workforce.

Apart from understanding healthcare providers' occupational routines with older patients, insight into their interactions is needed to design patient-centred care (PCC) for older adults that is aligned to health services' reality. Moreover, since healthcare needs to be increasingly coordinated and integrated to serve the needs of complex health issues and patient-centredness, a broader investigation comparing the variety of specialisations, including dental care, seems useful. Therefore, this study aimed to provide a deeper understanding of the perspectives of the variety of physicians and dentists providing ambulatory healthcare for people aged 80 and over, to understand the challenges they face and to draw conclusions about how healthcare delivery for older adults should be organized. The focus on this specific age group was chosen because of the already described increase in complex health issues and care needs around the age of 80 years, and also to deliver more nuanced insights since most studies on older people broadly summarize them in age categories such as 60+ or 65+ years [27,28]. The vast investigation of physicians and dentists was intended to meet the requirements of a necessary development of new interdisciplinary, integrative concepts of healthcare, explicitly incorporating the neglected area of oral health and care in older age [8,11,29–31].

Accordingly, three research questions guided the study:

1) How is care for people aged 80 and over characterized from the perspective of physicians and dentists?

2) How should the delivery of health and dental care for people aged 80 and over be designed from the perspective of physicians and dentists?

3) What influences the provision of good care for people aged 80 and over from the perspective of physicians and dentists?

## Methods

A qualitative online survey was conducted to answer the research questions. The study was approved by the Medical Ethics Review Board of the Medical Faculty at the University of Cologne (21–1152). Written consent by confirming an online tick box (mandatory to start the survey) was obtained. The study was pre-registered in the German Register for Clinical Trials (DRKS00024666). Since there is no reporting guideline specifically for qualitative surveys, reporting of the study is based on the consolidated criteria for reporting qualitative studies (COREQ) [32] and the consensus-based checklist for reporting of survey studies (CROSS) [33], as far as they were applicable to this research design. An overview of the methodological process is provided in Fig 1.

### Setting

The study was performed in Germany's largest federal state, North Rhine-Westphalia, and focused on the ambulatory healthcare sector. In Germany, ambulatory healthcare is separated, regarding organisation and reimbursement mode, from the hospital care and the rehabilitation

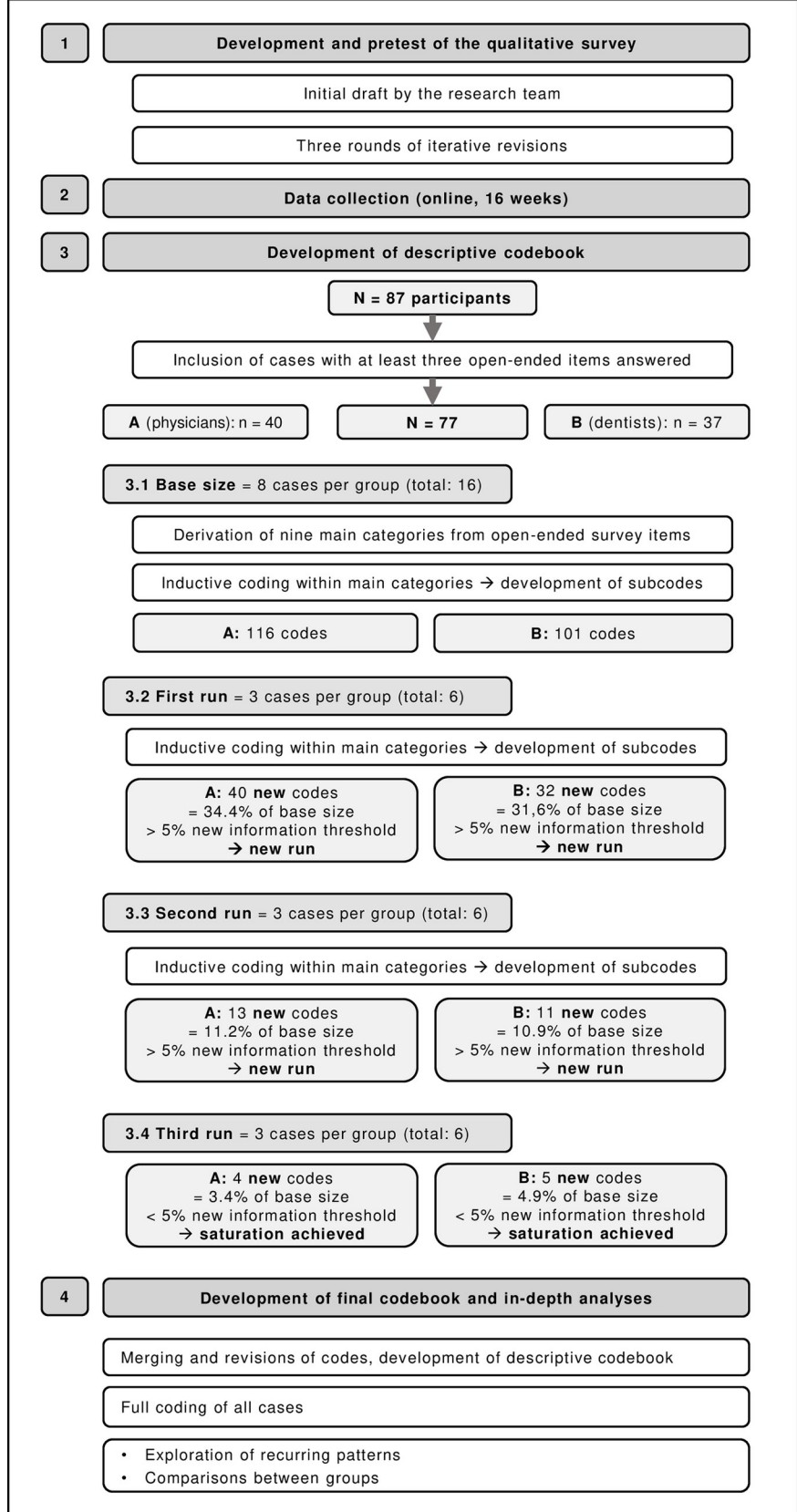

**Fig 1. Methodological overview.**

sector. GPs, specialists and dentists in the ambulatory healthcare sector provide services in their private practices [34]. Most of their services are covered by statutory health insurance for nearly 90% of the population; however, private co-payments for additional services are possible [34].

## Design and development of the qualitative survey

Employing a phenomenological approach, we chose a qualitative survey design, in a self-administered online mode, to answer the research questions. Apart from a few closed items, qualitative surveys primarily pose open questions following a qualitative research logic to explore the "participants' subjective experiences, narratives, practices, positionings and discourses" [35]. Qualitative surveys are useful in depicting the broader diversity of the phenomena of interest and in recruiting hard-to-reach groups; therefore, new and rich insights can be gained [35]. We chose this study design due to the breadth of the physicians' and dentists' specialisations and their usually limited availability for time-consuming qualitative studies due to their high workload. For this survey, they could flexibly choose from where and when they want to participate.

With regard to the research questions, relevant aspects and questions were brainstormed, initially. After discussion among the multidisciplinary research team (health sciences, health services research, medicine, dental care, sociology), the initial version of the survey was pretested. The pretest was based on iterative rounds of comments and revisions until the items worked as intended. This was reached after three rounds, the changes mainly concerning comprehensible wording of the items and the length of the survey. Details of the pretest rounds, the participants, and survey development are provided in an online (S1 Appendix).

The final survey contained nine closed items on sociodemographic and practice aspects. The eight open items asked for 1) characterization and challenges of working with patients aged 80 and over, 2) what constitutes good healthcare in older age from the healthcare providers' perspective, 3) influential factors (barriers and facilitators) of treating older patients as intended, 4) conceptions on ensuring that patients feel safe, feel like meaningful human beings and maintain control and independence (older people's core motives identified in [20]), and 5) proposals to improve or design future healthcare in old age. The complete survey is provided in the online S1 Appendix.

## Sampling and recruitment

Participants were eligible for the survey if

4) they were GPs or specialist physicians or dentists,

5) they practiced in an ambulatory healthcare setting in North Rhine-Westphalia, and

6) had sufficient reading and writing skills to answer a German questionnaire.

Physicians exclusively treating children and adolescents (e.g., paediatricians) or not practicing in an ambulatory healthcare setting in North Rhine-Westphalia were not eligible for participation.

We followed a convenience sampling strategy [36] to recruit physicians and dentists from diverse specialties but did not aim at statistical representativity. Because there are no clear guidelines for the sample size of a qualitative survey, we took the recommendations from Braun et al. as a reference point and aimed at recruiting 50–100 participants for a mid-range sample [35]. To reach the participants, 45 physician organizations (e.g., physician networks) were contacted and asked to promote the survey by spreading the survey link and a short

information text via newsletters, e-mail lists or their websites among their members. Fourteen organisations agreed to spread the survey.

## Data collection and analysis

The survey was created via SoSciSurvey. The participants were informed about all aspects of anonymous data processing and had to actively agree to participate to start the survey. The final dataset was transferred to MAXQDA Analytics Pro 2020 (Verbi software, Berlin). Cases were included for analysis if at least three open-ended items were answered.

The closed items were analysed descriptively. For the open items, we followed the approach of structuring qualitative content analysis designed by Kuckartz in developing a descriptive codebook [37,38]. Moreover, to increase transparency and assess whether data saturation could be reached within our sample, we decided to document the development of thematic saturation (little or no new information in additional cases regarding the research question) [39]. We set the base size at 16 cases (eight from Group A–physicians, and eight from Group B–dentists) for defining the initial body of information, and a run length of six cases (three per group) to define additional information. We set the new information threshold at $\leq 5\%$ (new themes found within a run compared to the base size) [39]. Therefore, new runs were analysed until the new information threshold was reached.

Following Kuckartz, the main categories were derived from the open items first. Second, two researchers (AH–health scientist, LV–sociologist, both female) independently read, memo-ed and inductively coded the first randomly chosen eight cases per group line-by-line and structured the results towards a first set of subcategories per main category. Both researchers discussed their results and agreed on a joint set of categories that was described in a codebook. Next, this was repeated with the first run of cases, in which newly identified codes were highlighted. After discussion, the codebook was revised accordingly. The procedure was repeated until the new information threshold was reached. Third, the codebook was reviewed and discussed by the research team. Subsequently, AH fully coded the remaining cases and the coding was checked by LV. The coded survey data was then examined regarding mutual patterns among the categories and compared between the two groups. The participants from the survey were not included in the data analysis process.

## Results

Data were collected from October 2021 to February 2022. In total, there were 87 participants. In n = 77 cases, at least three qualitative answers were provided and, therefore, they were included in the analysis. Both groups were comparable regarding demographics (Table 1). Most participants were aged between 50 and 60 years and male. Professional practice experience ranged from four to 44 years (mean 23 years). Nearly all the participants indicated they worked in an urban or mostly urban region. Most respondents stated that the proportion of patients aged 80 and over in their practice comprised up to 25%, and the vast majority had the perception of having received good education and training regarding caring for older adults. Approximately one third stated they were able to provide the care as desired in 75% or more of their cases, while 22 participants stated this applied to 49% or less of their cases (Table 2).

Thematic saturation was reached after three runs of independent code development. In the following, the results will be described in the broader themes identified throughout the different topics addressed in the survey: 1) characteristics of providing care for older people, 2) what matters to good healthcare in older age, 3) barriers and facilitators in the provision of good care. An overview of the topics is provided in Fig 2. The descriptions are based on the more detailed codes developed during analysis, which are depicted in Table 3. Moreover, the

**Table 1. Participant demographics.**

| | | Total | Physicians (n = 40) | Dentists (n = 37) |
|---|---|---|---|---|
| Age | 30–39 years | 9 | 3 | 6 |
| | 40–49 years | 6 | 2 | 4 |
| | 50–59 years | 31 | 18 | 13 |
| | 60–69 years | 27 | 14 | 13 |
| | 70–79 years | 4 | 3 | 1 |
| | missing | - | - | - |
| Sex | male | 51 | 27 | 24 |
| | female | 25 | 13 | 12 |
| | missing | 1 | 0 | 1 |
| Professional experience | mean/median (range) | 23/24 (4–44) | 21/22 (4–40) | 24/25 (5–44) |
| | missing | 2 | 2 | 0 |
| Specialization | general practice/internal medicine | | 15 | - |
| | internal medicine, geriatrics | | 1 | - |
| | internal medicine, 'no' or 'other' focus than geriatrics | | 5 | - |
| | surgery | | 1 | - |
| | otorhinolaryngology | | 3 | - |
| | neurology | | 3 | - |
| | nuclear medicine | | 1 | - |
| | physical and rehabilitative medicine | | 1 | - |
| | anesthesiology | | 1 | - |
| | urology | | 9 | - |
| | dentistry with focus on geriatric dentistry | | - | 7 |
| | dentistry with other focus of activity | | - | 9 |
| | dentistry without focus of activity | | - | 21 |

codebook, including example interview passages (German original with English translation), is provided in online S2 Appendix and the full coded material from the open items (German original) is provided in online S3 Appendix. Passages from interviews are labelled with GP for general practitioners, S for specialists and D for dentists.

## Characteristics of providing care for older people

**The complexity of multimorbidity and polypharmacy.** The participants, especially the physicians, referred to older patients' health status mainly as complex and limited and, consequently, challenging. In particular, they described multimorbidity and polypharmacy, causing more intensive care and time effort, limitations of mobility and care options, special attention regarding adverse events, and also the perception that older people themselves feel overstrained or helpless because of their medical complexity. It was further described that most disease-specific guidelines did not consider complex conditions such as multimorbidity or frailty and, therefore, were not useful in these situations. These circumstances were described as particularly challenging because it was often difficult to determine *"a reasonable scope of diagnostics and therapy"* (S-516, §5).

**Physical and cognitive restraints influencing treatment and communication.** Older adults' physical and cognitive limitations, often influencing interactions and treatments, were also frequently noted in both groups. Cognitive restrictions such as limited sensory perceptions (hearing, seeing), slowness and forgetfulness were explained as challenging for

**Table 2. Results of closed items.**

| | | Total | Physicians | Dentists |
|---|---|---|---|---|
| Estimated proportion of privately insured patients in the practice | <10% | 33 | 17 | 16 |
| | 11–25% | 34 | 19 | 15 |
| | 26–40% | 5 | 3 | 2 |
| | 41–65% | 4 | 0 | 4 |
| | 66–79% | 0 | 0 | 0 |
| | >80% | 1 | 1 | 0 |
| | missing | 0 | 0 | 0 |
| Estimated proportion of patients aged 80 and over in the practice | <10% | 16 | 7 | 9 |
| | 11–25% | 44 | 22 | 22 |
| | 26–40% | 10 | 6 | 4 |
| | 41–65% | 4 | 3 | 1 |
| | 66–79% | 3 | 2 | 1 |
| | >80% | 0 | 0 | 0 |
| | missing | 0 | 0 | 0 |
| Perception of having received good education and training | yes | 40 | 20 | 20 |
| | rather yes | 33 | 19 | 14 |
| | rather no | 4 | 1 | 3 |
| | no | 0 | 0 | 0 |
| | missing | 0 | 0 | 0 |
| Perception of providing good care (estimated proportion of all treated cases) | <25% | 8 | 2 | 6 |
| | 25–49% | 13 | 8 | 5 |
| | 50–75% | 30 | 16 | 14 |
| | >75% | 26 | 14 | 12 |
| | missing | 0 | 0 | 0 |

communication. In general, a dominant description was how communication with older patients was more demanding. This was explained by the limited capacity to understand and remember complex information and more questions, and a need for information: *"Difficulties in communication–patients absorb less information and cannot express themselves that well anymore"* (D-189, §17). Moreover, it was mentioned that communication also needed to cover psychosocial needs more strongly compared to younger age groups. Therefore, more time and adapted communication styles (e.g., reducing complexity, speaking clearly) were required. Additional time was also needed because of more complex treatment planning as well as to exchange information with other disciplines and stakeholders, according to both groups. Many of the participants believed they did not have the time to implement these approaches.

Physical limitations such as impaired mobility or declined motor skills were also described as demanding alternative concepts of treatment, e.g., including more breaks during the consultation or treatment. This was particularly reported by dentists, who could often not tilt older patients in the dental chair during treatment because of cardiovascular diseases, stiffness, and dizziness.

In contrast, six participants stated that older patients appeared to be heterogeneous, with some still being fit and self-determined and not causing any additional effort.

**Unclear or conflictual care planning.** In both groups, the participants described that some older people accepted their health status as normal due to their age and that they held the view or were unsure whether treatment would still be worth it. Conversely, some physicians stated that older people held unrealistic views of their health status and the possibilities

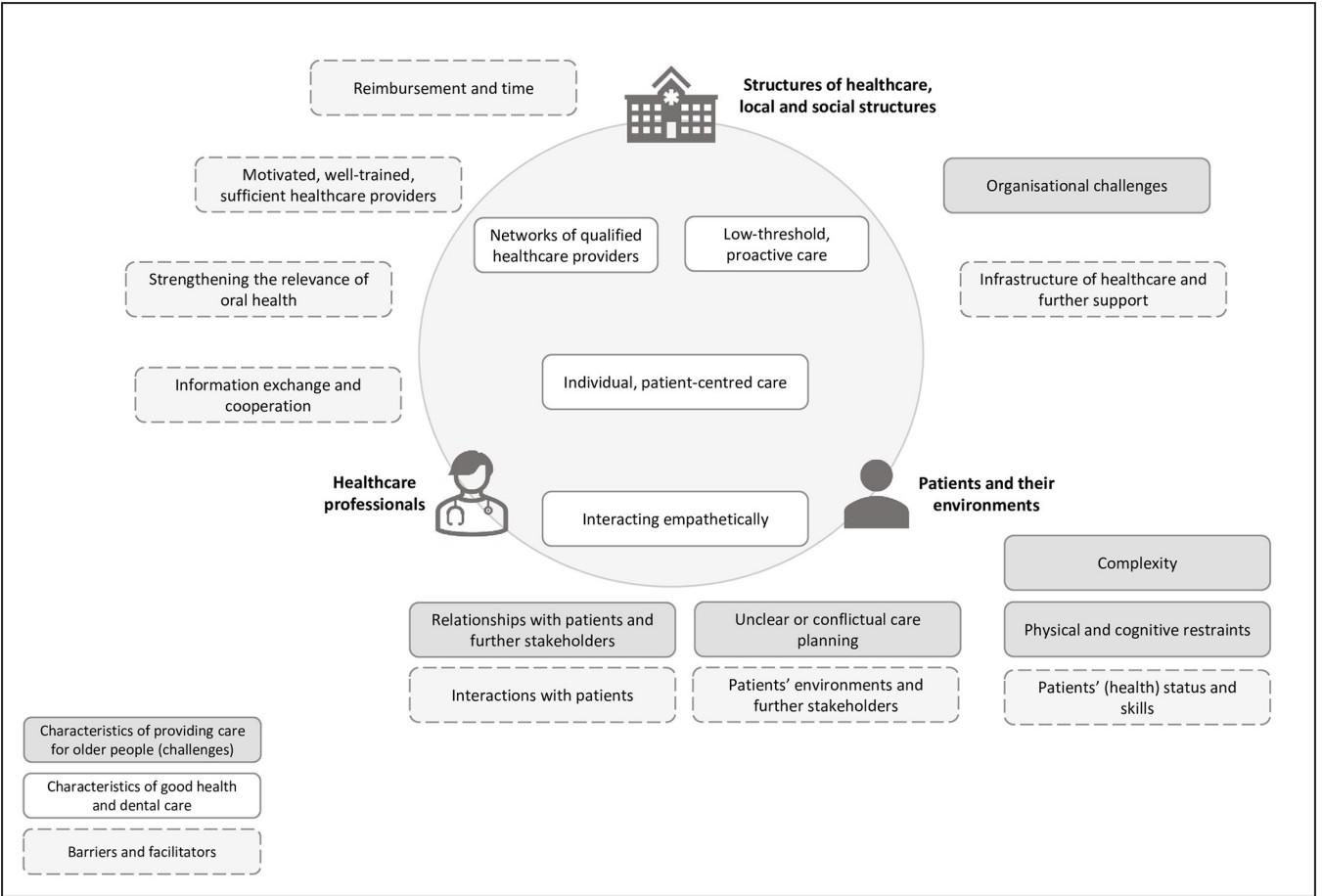

**Fig 2. Overview of results.**

of diagnostics and treatment. Concomitantly, conflicting views on older people's compliance were outlined. On the one hand, especially GPs had the perception that older people were more paternalistic-oriented and more compliant. On the other hand, a lack of compliance and resistance to advice were also described, sometimes characterized as *"stubbornness"* (S-217, §10). Regardless of the cause, it was described that non-compliance increased physicians' and dentists' expenses.

Older patients' care goals and non-medical concerns were perceived as challenging; this was most dominantly described by GPs. The first challenge on this matter was that reasonable treatment was not always clear, nor were patients' individual goals, which could differ from the physician's or dentist's views. Moreover, psychosocial concerns such as loneliness, lack of familial or social support, financial problems and the organisation of support were described as additional demands on healthcare. The participants found themselves confronted with these demands, which could be of relevance in providing a good care plan, but likewise, several did not perceive it as their area of competence or could not provide the resources necessary for fulfilling them, e.g., due to limited time.

**Organisational challenges.** As a result of the more complex care situation, treatment concepts were described as being more individualized and frequently incorporating *"the solution of daily problems and organisational regards"* (GP-253, §5). Sometimes, the organisation of social or specialist services was considered a part of the treatment concept. The dentists also

**Table 3. Overview of descriptive codebook and code frequencies.**

| Main category (number of cases answering) | Subcategories/codes | No. of cases coded among physicians* | No. of cases coded among dentists* |
|---|---|---|---|
| Characteristics of working with people aged 80 and over (n = 76) | Role of life and experiences | 2 | 1 |
| | Positive attributes | 8 | 6 |
| | Challenging attributes | 15 | 4 |
| | Complexity due to multimorbidity and polypharmacy | 12 | 7 |
| | Physical and cognitive limitations | 19 | 20 |
| | Heterogenous appearance | 3 | 3 |
| | Higher need for care | 0 | 3 |
| | Perspective of people aged 80 and over regarding their health and care | 7 | 6 |
| | Paternalistic orientation | 4 | 0 |
| | Discernment and compliance | 3 | 4 |
| | Relationship-building and trust | 4 | 3 |
| | Speed and time | 20 | 11 |
| | Continuity and control | 2 | 2 |
| | Need for support | 2 | 1 |
| | Additional stakeholders and actors | 7 | 6 |
| | Communication | 9 | 10 |
| | Treatment concepts | 6 | 11 |
| | Structural and organizational specific features | 4 | 5 |
| Good healthcare in old age (n = 75) | Individual, person-centred view and care | 15 | 10 |
| | Status and behaviour of patients | 3 | 2 |
| | Patient-relevant outcomes | 12 | 12 |
| | Empathy and appreciation | 4 | 4 |
| | Communication | 6 | 2 |
| | Time | 11 | 7 |
| | Proactive care | 9 | 18 |
| | Access and infrastructure | 13 | 10 |
| | Sufficient and well-trained staff | 3 | 5 |
| | Cooperation with further actors | 5 | 5 |
| | Patients' environments | 5 | 2 |
| Challenges in caring for people aged 80 and over (n = 73) | Complexity due to multimorbidity and polypharmacy | 13 | 7 |
| | Influence of physical and cognitive limitations | 14 | 15 |
| | Lack of compliance or rejection of treatment | 8 | 4 |
| | Handling of time resources | 10 | 7 |
| | Relatives and further actors | 3 | 10 |
| | Patients' care goals and (non-medical) further issues | 10 | 3 |
| | Structural and organizational challenges | 9 | 9 |
| | No challenges | 1 | 1 |
| Reasons for not providing the desired care (n = 72) | Patients' (health) status and abilities | 7 | 22 |
| | Patients' environments and further actors | 11 | 7 |
| | Lack of compliance or discernment | 14 | 9 |
| | Financial Reasons | 2 | 16 |
| | Expenditure/lack of time | 11 | 3 |
| | Lack of information exchange and cooperation with other actors and disciplines | 4 | 5 |
| | Lack of (qualified) staff | 5 | 1 |
| | Bureaucracy | 2 | 1 |
| | Design of healthcare infrastructure | 8 | 7 |

*(Continued)*

**Table 3.** (Continued)

| Main category (number of cases answering) | Subcategories/codes | No. of cases coded among physicians* | No. of cases coded among dentists* |
|---|---|---|---|
| Facilitators of providing good care (n = 70) | None | 0 | 1 |
| | Care providers' experiences | 5 | 3 |
| | Care providers' qualifications and training | 7 | 6 |
| | Care providers' attitudes and motivation | 19 | 13 |
| | Encounters with patients | 13 | 4 |
| | Patients' (health) status | 2 | 2 |
| | Interactions with further stakeholders | 12 | 8 |
| | Time | 4 | 1 |
| | Interdisciplinary cooperation | 7 | 1 |
| | Local structures and offers | 4 | 1 |
| | Supporting tools and programmes | 4 | 0 |
| Ensuring a feeling of safety (n = 71) | Not possible | 1 | 0 |
| | Familiar environment | 2 | 2 |
| | Timely contact options | 4 | 1 |
| | Proactive care | 2 | 3 |
| | Support | 5 | 6 |
| | Good communication, counselling and conversations | 17 | 7 |
| | Attention and appreciative behaviour, relationship-building | 12 | 19 |
| | Sufficient time | 1 | 5 |
| | Competence and education | 2 | 3 |
| Ensuring a feeling like that of a meaningful human being (n = 72) | Not possible | 1 | 0 |
| | Good communication, counselling and conversations | 4 | 7 |
| | Attention and appreciative behaviour, relationship-building | 25 | 19 |
| | Sufficient time | 9 | 6 |
| | Engagement with individual needs and wishes, holistic view | 8 | 9 |
| | Social integration | 4 | 0 |
| Ensuring the maintenance of control and independence (n = 65) | Not (always) possible | 3 | 1 |
| | Engagement with individual needs and wishes, holistic view | 1 | 2 |
| | Good communication and counselling | 5 | 3 |
| | Appreciative behaviour | 1 | 3 |
| | Enabling decision options | 6 | 6 |
| | Encouragements and support without paternalism | 5 | 5 |
| | Maintaining and expanding functionalities, prevention | 11 | 1 |
| | Structures of support and care | 15 | 5 |
| Measures to improve healthcare in old age (n = 67) | Prevention orientation | 3 | 4 |
| | Patient orientation | 6 | 2 |
| | Empathy | 2 | 2 |
| | Reimbursement | 16 | 12 |
| | More time | 10 | 4 |
| | Access | 2 | 4 |
| | Support structures | 15 | 10 |
| | Information exchange and interdisciplinary cooperation | 5 | 4 |
| | Simplification | 0 | 2 |
| | Qualification and training | 3 | 7 |
| | More (qualified) staff | 5 | 3 |
| | Consideration of non-medical dimensions | 8 | 1 |
| | Strengthening the relevance of oral health | 0 | 14 |

* The frequencies are shown to provide more transparency and insight into the findings but are not appropriate for drawing conclusions based on their weighting or importance.

highlighted how they focused their treatment concepts on reduced therapy, uncomplicated handling and, particularly, on the absence of pain.

Moreover, participants in both groups described that for older patients, more effort regarding the organization of (social) support and transport had to be made. One of the most important challenges was that necessary ambulatory and local support structures decreased and were more cumbersome to obtain. Specialists and dentists also complained about a lack of information and the necessary equipment to adequately treat older patients. The lack of information was further explained by missing medical reports and information from other disciplines. Moreover, care at home was perceived to be difficult to ensure due to bureaucratic and financial hurdles, especially when *"the financial means of the older person are depleted"* (GP-525, §8). Dentists also pointed out that providing dental care was difficult because staff and equipment needed to be transported and cooperation with the nursing home staff was sometimes bad.

**Relationships with patients and further stakeholders.** Some participants stated that caring for older adults often means building relationships over many years, resulting in deep trust. This was also mentioned as a demand of the older patient group, which was described to be *"happy to receive personal attention, a conversation away from the condition"* (GP-198, §3) and sometimes needed special sensitivity.

In both groups, it was noted that the entire care situation changes when relatives or caregivers accompany the older patient. This could be perceived as helpful, but they were also stakeholders with their ideas and demands, resulting in additional expenditure of time. This was explained by sometimes differing own interests and goals of these people, as well as *"utopian demands"* (D-374, §3). It was also found that consultations with relatives and formal caregivers could be complicated and bureaucratic, e.g., regarding cost coverage of necessary treatments such as dentures.

## Good healthcare in older age–what matters?

**Treating a person, not a case: Individual, patient-centred care.** Providing an individual, patient-centred view and treatment was a dominant feature of good healthcare in older age for both groups and considered very important when addressing older patients' core motives. This was described as ensuring *"individually adjusted therapy concepts, less guideline medicine"* (S-232, §4) and adapting the care plan to the patients' individual wishes, circumstances, functionalities, and treatment goals. The most important goal explained in this regard was maintaining independence and autonomy as best as possible, including mobility and functions such as chewing. Moreover, maintaining quality of life, guaranteeing freedom from pain, and providing prevention and control-oriented care were also found to be important. Balancing risks and benefits-oriented towards patients' functionalities and feasibility were described, as was the necessary courage to provide less treatment and medication. Moreover, focusing on mental health and well-being as well as inquiring about the patient and listening well ("narrative medicine") were mentioned.

Motivating patients to maintain activities and preventive measures, and to use early interventions if functions were declining were highlighted to support their maintenance of control and independence, as well as encouraging them to accept support, to engage in improving functions and recommending supportive measures without paternalism. Pointing out different treatment options and providing good counselling to them and, therefore, enabling the scope for decision-making were reported to strengthen patients' self-esteem and autonomy.

**Interacting empathetically without time pressure.** For a range of cases in both groups, communication was important in providing good healthcare. This was primarily described as

listening patiently, paying attention, and addressing questions and concerns comprehensively to provide good counselling and comprehensible explanations. Therefore, "*granting sufficient time*" (D-454, §12) and acting empathetic, appreciative, respectful, and friendly were considered necessary. In both groups, taking sufficient time for the interaction was particularly considered to make the patients feel safe and meaningful. Good communication, counselling and conversations were also among the most important aspects to address all three older patients' core motives and build trustful relationships, incorporating interactions on an equal level and respect for the older patients' life's work.

In contrast, three GPs and two dentists also mentioned that the interaction and therefore, healthcare, is good when patients are compliant and have realistic expectations.

**Providing low-threshold, proactive care.** A very important feature in both groups, but disproportionally often described by dentists, was proactive care. Proactive care was described as providing reliable and continuous examinations and check-ups, strongly connected with broader offers of prevention (e.g., regarding mobility and dental prophylaxis) and dental hygiene (possibly with help) to maintain independence. Moreover, involving relevant people such as relatives or caregivers in planning care was considered proactive care. Some dentists additionally referred to enabling manageable care as well as long-lasting prostheses.

Access and infrastructure were also a feature of good healthcare in both groups, but especially dominant among the specialists. This aspect included medical and dental care located nearby or, alternatively, the availability of low-threshold opportunities for transport to reach a practice. Moreover, access and provision of further offers such as "*social assistance, networks, ambulatory support*" (S-470, §4) were described regarding this. Especially, physicians referred to ensuring these structures to address older patients' motives to maintain control and independence, including the accessibility of information and optimisation of living and care facilities. Furthermore, ensuring support regarding medication intake, the involvement of relatives, and enabling the use of aids and restorative measures, were suggested to help address older patients' core motives.

Another important aspect was low-threshold access to necessary therapies and diagnostics, as well as preventive examinations, without bureaucratic or financial barriers (e.g., cost coverage by insurances): "*The semi-annual check-up is important because motor skills decline and the oral hygiene is not ensured sufficiently anymore*" (D-439, §6). Additionally, timeliness and continuity of care, sufficient (specialist) physicians and accessibility were mentioned.

**Networks of qualified healthcare providers.** Another aspect discussed under good healthcare was the availability of sufficient well-trained staff. Participants referred to broadly trained professionals, who are aware of the necessary aspects of caring for older patients. They also explained that healthcare providers' perceived competence would make the patients feel safe. Moreover, the "*close cooperation of all professional actors*" (S-516, §4), such as medical professionals, nurses, caregivers, and social services, was proposed to enable holistic care and comprehensive, individually shaped support for patients. Three dentists also mentioned cooperation with nursing homes to provide good oral healthcare and hygiene.

## Barriers and facilitators in the provision of good care

**Patients' (health) status and skills.** Older patients having bad overall (health) was described as a barrier to providing care as desired, predominantly by dentists. It was held that physical limitations, namely decreased resilience, bone loss and limited motoric skills, made the desired treatments difficult to impossible. Moreover, multimorbidity and polypharmacy leading to medical complexity and an unstable overall health status hampered the desired treatments. Concomitantly, good overall health and functionality in the patients were described as facilitators by some participants.

A lack of skills and functionalities necessary to master daily life, as well as private or social barriers regarding support or financial resources, were also discussed as ultimately negatively influencing patients' health status and, therefore, medical or dental treatment. Additionally, bad cognitive functioning and dementia were mentioned by some participants as barriers to cooperation. Consequently, a stronger prevention orientation was described as necessary: *"make provisions at a young age, to need medical help as little as possible in old age"* (S-499, §13). While physicians focused on cultivating a health-promotive lifestyle early in life, dentists rather focused on offers and coverage of dental prophylaxis and preventive routine examinations.

**Patients' environments and further stakeholders.** Patients' environments, including further stakeholders such as relatives, were also mentioned as possible barriers to providing the desired care, especially by GPs. A lack of acceptance of patients' wishes, lack of support or discernment by relatives and formal caregivers, as well as *"a too high and counterproductive sense of entitlement"* (GP-245, §7) were described as impediments to good care. However, the involvement and support of relatives and caregivers were mentioned as a possible facilitator as well. This could be achieved by meaningful exchanges with them and additional support to make good decisions for the patient.

**Interactions with patients.** In both groups, but most dominantly by GPs, meaningful interactions with the older patients were mentioned as facilitators. *"Intensive conversations"* (GP-202, §19) and patients' gratitude were perceived as being supportive. Moreover, the patients' cooperation and their stable social environments were also described as supportive. Consequently, focusing more on the patients as individuals and granting them dignified and respectful care were described as necessary measures to improve healthcare. On this matter, narrative medicine was highlighted as a principle that should be supported by the organisation and reimbursement of the healthcare system.

However, a lack of patient compliance was described as hindering the desired care, especially by GPs. Physicians explained this factor as patients lacking discernment in the necessity of treatments, stubbornness and not adhering to treatment plans. Some dentists described that there might be discrepancies between the patients' wishes and their perception of adequate treatment and, therefore, the latter would not be provided.

**Reimbursement and time.** A lack of time to care for and interact with older patients as desired was mentioned in both groups, more dominantly by physicians, and especially against the background of greater demand on time, as already described. Consequently, the most important measure suggested was the reorganization of reimbursement: namely, providing higher renumeration for health services in general or, in particular, for older patients. In general, it was described that it would be desirable to receive more renumeration for greater efforts regarding time spent and to enable narrative medicine, which would otherwise be hardly or not possible at all. Moreover, cost coverage for further or necessary therapies and treatments such as prophylaxis, oral hygiene, physiotherapy or occupational therapy were found necessary. Especially dentists described how desired care depended on the *"patients' [limited] financial resources"* (D-399, §8).

**Information exchange and cooperation.** Specialists noted that one reason for not providing the desired care was *"a lack of information and cooperation"* (S-516, §7) and the absence of complete medical records to adequately treat their patients. Moreover, the specialists and dentists explained that cooperation with other disciplines and actors was needed to provide comprehensive and good care, but that their availability was often limited, communication was lacking, and that they also perceived a lack of willingness to cooperate. In particular, the specialists valued the exchange with colleagues and cooperation with other disciplines involved in the care of the older patient as a facilitator, as well as having a professional network. In both

groups, better interdisciplinary cooperation and networking among all care providers were explicitly requested.

**Motivated, well-trained, and sufficient healthcare providers.** One of the most important facilitators described in both groups was the attitude and motivation of medical professionals. To provide good care despite the described obstacles, it was explained that idealism and engagement, as well as empathy and a strong moral claim to supporting older patients and guaranteeing them the healthcare they deserve were facilitating. It was also mentioned that the participants thought of their own future as being old or *"my older parents"* (GP-198, §10) and the care they would like to receive. Moreover, the motivation, the attitude towards older patients and *"routinely well-organized team[s]"* (D-384, §8) were mentioned as important facilitators, as well as their professional experience in providing health or dental care for such patients. On-going qualification and training regarding the specialties of caring for older patients was additionally described as being supportive in providing good care. However, in both groups, the participants mentioned a lack of qualified staff as a barrier, and several suggested the recruitment and education of further qualified staff.

**Strengthening the relevance of oral health.** Several dentists highlighted the necessity to raise awareness of the relevance of oral health in older adults among relatives, physicians and further actors. This included the integration of oral health matters and provision or support of oral hygiene, especially in care-dependent older adults, in education and training of medical staff, nurses and relatives. Moreover, broader possibilities and the cost coverage of prophylaxis in old age and support for visiting dental care were demanded. It was stated that *"the dental care in nursing homes has been structurally secured in the past years–the ambulatory care still is a broad field"* (D-451, §25–26). Finally, dedicated dental care rooms for visiting dentists in nursing homes were suggested.

**Infrastructure of healthcare and further support.** In both groups, the design of healthcare structures was discussed as an important factor in providing the desired care. This concerned a lack of cost coverage by insurance schemes, lacking therapy places, and *"age-appropriate transportation systems"* (S-175, §8). Moreover, a lack of possibilities to provide home and nursing home visits was mentioned. Improvement of access was described as necessary, especially regarding cost coverage and the simplification of application for treatment and therapies, home visits and preventive appointments.

Moreover, low-threshold opportunities for support structures such as ambulatory services, *"patient guides that take organisational tasks"* (GP-158, §16) or welfare centres were suggested. Provision of barrier-free information, help in mobility and transport, daycare centres and visiting or mobile health and dental care were said to be necessary. Therefore, a good local infrastructure of social services and ambulatory care services were said to be facilitators of good healthcare. Support structures, such as local networks guaranteeing social integration and participation in cultural activities to prevent loneliness, were also required to serve older patients' psychosocial well-being.

## Discussion

This qualitative survey investigated the views of physicians and dentists on caring for people aged 80 and over in the fields of ambulatory health and dental care. We comparatively described how physicians and dentists characterize these care interactions, what they perceive to be good healthcare in older age, and which factors influenced this. The identified themes are closely interrelated, evidenced by certain factors occurring frequently in various roles. For example, "more time needed" is a characteristic of caring for older adults, while "too little time" is a challenge and a barrier to providing good care. Meanwhile, "having sufficient time" is a characteristic of and facilitator for good care.

Overall, our results show that caring for people aged 80 and over is challenging for physicians and dentists due to inherent complexity, a decline in health status, or physical and cognitive limitations that influence treatment possibilities and communication. Moreover, the entire care situation seems to have changed in caring for older adults due to 1) the mentioned complexity and, therefore, less obvious or fewer possible treatment options, 2) the older patients' genuine view on the desirability and value of treatments, 3) the greater relevance of the patients' social and local surroundings since support is needed, and 4) further stakeholders, such as relatives or caregivers, who need to be considered in care interactions and who often bring their interests and demands.

These findings are in line with a study by Zwisen et al. investigating GPs' views on complex older patients in the Netherlands [17]. Caring for complex or older patients means that direct and established strategies for providing care, as are also depicted in guidelines for single conditions or the certainty of being only in charge of a medical issue, do not work properly. This is also mirrored in the frequent complaints regarding older patients' lack of compliance or the refusal of treatment in our survey–something does not work as intended, and this puts stress on the healthcare providers.

For both physicians and dentists, good care for older people was characterized by the provision of individualized concepts, empathetic interactions and low-threshold, proactive care in a network of qualified healthcare providers. Therefore, their perceptions are in line with those of older patients, [40,41] and conceptualisations of PCC [42,43]. However, this ideal is currently hampered by insufficient time and unhelpful reimbursement structures, and the fragmentary infrastructure of healthcare and further social services. This is supported by the considerable proportion of participants stating that they were able to provide the care they found appropriate in 49% or less of their cases. In a study that externally rated PCC in consultations, it has also be shown that PCC is currently moderately apparent [44].

Consequently, the wish and the demand to provide PCC seem to be given, but this stands in contrast to the limited resources and opportunities, possibly resulting in the reported feelings of overload or frustration. Therefore, the assumption that healthcare providers primarily stem the challenges of the demographic shift in healthcare while the healthcare system does not provide adequate support seems confirmed. According to a study among GPs by Herzog et al., there are three ways GPs can manage these demands and this complexity: 1) by focusing on medical expertise and not seeing themselves in the position to serve all demands; 2) by holding a holistic view, trying to serve all demands using great individual effort; 3) by seeing oneself as one part of a broader (social) network that overall serves the demands [21]. The last strategy seems to be the most promising for physicians and dentists in the light of our results since the most frequent suggestion, besides improved reimbursement and more time, was the development and low-threshold access to further support structures, including social or welfare services.

Therefore, future efforts to design PCC suitable for the older population should focus on creating broad networks of a variety of medical providers, but not being limited to them. It was also a striking finding that relevant healthcare areas, such as dentistry, are widely unrecognized according to the dentists in our survey, who frequently complained about a lack of awareness regarding oral health and hygiene among healthcare providers, patients and relatives–resulting in a bad oral status. This finding is in line with several other studies [45–49]. Moreover, while care contacts among older people have increased in most areas of ambulatory healthcare due to increased demand, care contacts in dentistry have declined in patients in older age [50]. Therefore, the dentists demanded an increase in awareness and, in particular, a prevention-oriented care organisation.

In contrast to the current healthcare structures, which were mainly described as hindering, the most important facilitators described by physicians and dentists were their own and their staff's motivation, a positive attitude towards older people, and sufficient professional experience. While training and early exposure are frequently suggested measures [21,51], the participants in this survey valued training as a facilitating factor, even though they stated they felt well-educated themselves. Therefore, courses addressing the handling of complexity and focusing on motivation and attitudes regarding caring for older adults might be more promising than knowledge-based education. This is supported by studies among nursing and medical students which found that correcting false beliefs about older people, focusing on empathy-building and enhancing a better understanding of the ageing process made them more positive about and confident in treating older patients [16,18,19,52–55]. To support the building of care networks and strengthen a comprehensive understanding of the very old patient, broadly including the relevant areas for maintaining autonomy and well-being, this training could be provided in an interdisciplinary setting.

## Possible influence of the COVID-19 pandemic on study results

This study was conducted between October 2021 and February 2022, thus, during the on-going COVID-19 pandemic. Studies show that in Germany, especially during the first phase of the pandemic in the beginning of 2020, the legislative protection measures affected healthcare utilization. Approximately one third of a representative sample of the general population stated to have a medical, dental or therapeutical appointment postponed or cancelled (by themselves or the providers) [56]. This mostly regarded preventive and routine examinations, especially in dental care [56,57]. However, decreases were especially seen among people aged 35 years or younger and in the field of GP care, while specialist consultations, particularly among people aged 75 or 80 and over, remained stable [56–58]. In the course of 2021, a general stabilization of ambulatory healthcare utilization occurred, despite on-going or relaunched legislative protection measures [58,59]. Apart from the partially volatile patient numbers, healthcare providers were facing an additional burden especially due to diagnosis and treatment of COVID-19 in ambulatory healthcare practices, increasing alternative modes of consultation such as telephoning, infection protection measures in the practice and increased social tensions or deteriorated working climate [57,59,60]. Hence, what do the pandemic circumstances mean for the study in hand? Initially, we aimed to pose a question in the end of the survey to investigate whether the participants themselves had the impression their experiences or views had changed during the pandemic: *Reflecting briefly on your previous answers, has the corona pandemic changed your views and if so, how?* However, based on our own previous experiences with such questions in other qualitative studies during the pandemic [61,62] and also no meaningful results during the pretesting, we decided to delete the item after pretest round 2 (see S1 Appendix). This was also decided due to the survey questions being kept in a general manner and under the impression that the pandemic situation in the end of 2021 had started to normalize, which is at least partly supported by the available studies on ambulatory healthcare. To see whether the participants themselves raised any pandemic-related topic, we cross-searched the data set and only found three statements: 1) a GP reporting that in "normal" (non-pandemic) times, a practice assistant is supportive in caring for older people, 2) a dentist reporting that due to the pandemic, access to nursing homes became difficult, and 3) another GP reporting that due to the discourse on vaccinations, there was more distrust in the doctor-patient-relationship (see pages 9, 36, 51 in S3 Appendix). Hence, the pandemic did not seem to play a considerable role for this survey, despite potentially more persons might have participated if their workload was lower. However, it is possible that the participants' views are

under the impression of the overall additional burden and a general discourse on older people's vulnerability in the pandemic.

## Strengths and limitations

To the best of our knowledge, this is the first qualitative study investigating the various medical disciplines in the field of ambulatory healthcare for very old people. The results provide a comprehensive insight into the structures and disciplines that are expected to interact in providing good health and dental care. The use of a qualitative survey was a useful data collection method since it is open and flexible enough to gain qualitative insights, but also reaches a broader target group and, therefore, exploring greater heterogeneity is possible [35]. Another strength of the study is the thorough pretesting to ensure that the unobserved data collection worked well [35], and the development of codes using transparent reporting on the development of saturation to allow for an appraisal, whether sufficient reports could be collected or not.

However, some limitations should be kept in mind. First, despite the broad target group, not all the specialities of the medical field were represented in the sample, the sample mainly consisted of male participants aged 50 and older with considerate experience in practice, and the sample was located in an urban or mostly urban environment. This limits the transferability of findings to other contexts or groups. Moreover, due to the sampling strategy, possibly only people who were interested in the topic or who had strong opinions were willing to participate. Since the recruitment strategy was not personalized and used broad ways such as advertisement on the recruiting organisations' websites, it could not be assessed how many eligible persons were reached, and consequently, no response rate could be determined. Additionally, it was theoretically possible to participate more than once. However, this seems unlikely due to the qualitative nature of the study, and since the answers were not weighed and the heterogeneity was probably greater in this survey than in other, smaller sampled types of qualitative studies, the range of views still appears to be a reasonable representation.

Second, the qualitative survey method does not allow for interactive inquiries. Therefore, the results should be seen as descriptive, with explanatory approaches, rather than as solidly theory-generating. In future studies, a combined approach, e.g., a qualitative survey followed by a small number of additional interviews to provide more depth in explanatory approaches, could be promising.

## Conclusion

Providing ambulatory health and dental care for people aged 80 and over poses special challenges to physicians and dentists. There is an apparent willingness to provide PCC as a means of delivering good healthcare to older people, but doctors feel overstrained by the patients' demands and limited in their scope of action by an ill-equipped system. Efforts to reorganize the structures should focus on building networks with low-threshold access to different providers serving older patients' demands, and education for healthcare professionals should focus on handling complexity and creating positive attitudes towards the patients. Moreover, awareness of older people's oral health and a broader offer and coverage in providing dental care and hygiene in this age groups need to be strengthened to realize comprehensive care.

## Supporting information

**S1 Appendix. Survey development, pretest report and final survey.**
(PDF)

**S2 Appendix. Full codebook with example interview passages.**
(PDF)

**S3 Appendix. Full coded material from the open items.**
(PDF)

## Acknowledgments

We would like to thank our colleagues and supervisors at the Graduate School GROW for their valuable input into this paper and our student assistant Lina Beck for her support with formatting the manuscript. Moreover, we are indebted to all participants.

## Author Contributions

**Conceptualization:** Angélique Herrler, Vera Vennedey, Stephanie Stock.

**Formal analysis:** Angélique Herrler, Lisa Valerius.

**Investigation:** Angélique Herrler, Lisa Valerius.

**Methodology:** Angélique Herrler, Anna Greta Barbe, Vera Vennedey, Stephanie Stock.

**Project administration:** Angélique Herrler.

**Supervision:** Stephanie Stock.

**Visualization:** Angélique Herrler.

**Writing – original draft:** Angélique Herrler.

**Writing – review & editing:** Angélique Herrler, Lisa Valerius, Anna Greta Barbe, Vera Vennedey, Stephanie Stock.

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
