## [Decision Letter · Decision Letter 0]

14 Jul 2022

PONE-D-22-11053Providing ambulatory healthcare for people aged 80 and over: views and perspectives of physicians and dentists from a qualitative surveyPLOS ONE

Dear Dr. Herrler,

Thank you for submitting your manuscript to PLOS ONE. After careful consideration, we feel that it has merit but does not fully meet PLOS ONE’s publication criteria as it currently stands. Therefore, we invite you to submit a revised version of the manuscript that addresses the points raised during the review process.

We look forward to receiving your revised manuscript.

Kind regards,

Dylan A Mordaunt, MD, MPH, FRACP

Academic Editor

PLOS ONE

Journal Requirements:

Additional Editor Comments:

Thank you for your submission. This is written well and reads well. My main question is the study context including the reason for undertaking the study, age-group and study design, weren't entirely clear.

With regards to the criteria for publication:

1. The study appears to present the results of original research.

2. Results reported do not appear to have been published elsewhere.

3. Experiments, statistics, and other analyses are performed to a high technical standard and are described in sufficient detail, excepting the study question and context could be refined.

4. Conclusions are presented in an appropriate fashion and are supported by the data.

5. The article is presented in an intelligible fashion and is written in standard English.

6. The research appears to meet all applicable standards for the ethics of experimentation and research integrity.

7. The article adheres to appropriate reporting guidelines and community standards for data availability. I think the structure is clear and likely conforms to the main requirements. For studies that clearly meet a reporting format, I always suggest that the authors double-check conformance with that checklist, in this case it would be CROSS (https://www.equator-network.org/reporting-guidelines/a-consensus-based-checklist-for-reporting-of-survey-studies-cross/).

Reviewers' comments:

Reviewer's Responses to Questions

**Comments to the Author**

1. Is the manuscript technically sound, and do the data support the conclusions?

Reviewer #1: Yes

2. Has the statistical analysis been performed appropriately and rigorously? 

Reviewer #1: Yes

3. Have the authors made all data underlying the findings in their manuscript fully available?

Reviewer #1: Yes

4. Is the manuscript presented in an intelligible fashion and written in standard English?

Reviewer #1: Yes

5. Review Comments to the Author

Reviewer #1: Manuscript by Herrler et al is well written. It’s an important survey which explains what outpatient healthcare systems require to provide proper care to patients aged 80 and over. I have few concerns before manuscript accepted for publication.

Abstract: Line 28, how many ambulatory healthcare sectors were included in the survey. It would reflect the global scenario of the survey being held.

Line 34; Author mentioned, ‘Caring for patients aged 80 and over was perceived as challenging because of its complexity…’ Are authors talking about the complexity of disease? Please elaborate.

This survey was carried out in year 2021-2022. COVID situation must influence overall outpatient resources and care facility. Author could discuss this aspect in discussion section.

Format of references should be consistent. Check references 28, 31, 45

All images are blurred. High quality images are required

6. PLOS authors have the option to publish the peer review history of their article (what does this mean?). If published, this will include your full peer review and any attached files.

Reviewer #1: No

---

## [Author Response · Author response to Decision Letter 0]

21 Jul 2022

Comments on journal requirements

--

We checked and ensured the style requirements.

--

--

Participants had to confirm a mandatory online tick box regarding their consent to start the survey. We provided additional details in the methods section of the manuscript (p. 6, lines 115-116) and in the online submission information.

--

3. In your Data Availability statement, you have not specified where the minimal data set underlying the results described in your manuscript can be found. PLOS defines a study's minimal data set as the underlying data used to reach the conclusions drawn in the manuscript and any additional data required to replicate the reported study findings in their entirety. All PLOS journals require that the minimal data set be made fully available. For more information

about our data policy, please see http://journals.plos.org/plosone/s/data-availability. Upon re-submitting your revised manuscript, please upload your study’s minimal underlying data set as either Supporting Information files or to a stable, public repository and include the relevant URLs, DOIs, or accession numbers within your revised cover letter. For a list of acceptable repositories, please see http://journals.plos.org/plosone/s/data-availability#loc-recommended-repositories. Any potentially identifying patient information must be fully anonymized. Important: If there are ethical or legal restrictions to sharing your data publicly, please explain these restrictions in detail. Please see our guidelines for more information on what we consider unacceptable restrictions to publicly sharing data: http://journals.plos.org/plosone/s/data-availability#loc-unacceptable-data-access-restrictions. Note that it is not acceptable for the authors to be the sole named individuals responsible for ensuring data access. We will update your Data Availability statement to reflect the information you provide in your cover letter.

--

All relevant data are within the manuscript and its supporting information files (especially S3). Since the data are of qualitative nature, we provided the material only in its coded form and without sociodemographic information on case-level to ensure anonymity. We specified the respective information in the data availability statement.

--

4. We note that the grant information you provided in the ‘Funding Information’ and ‘Financial Disclosure’ sections do not match. When you resubmit, please ensure that you provide the correct grant numbers for the awards you received for your study in the ‘Funding Information’ section.

--

The study itself did not receive specific funding. All work was conducted within a graduate school on gerontological research, that itself was funded by the Ministry of Culture and Research of the State of North-Rhine Westphalia. However, the graduate school has no grant number and the ministry was not involved in planning or conducting the study. We added additional information in this section to provide a better explanation of these circumstances.

--

--

We checked and ensured the correctness of the reference list.

--

Comments of the academic editor

1. Thank you for your submission. This is written well and reads well. My main question is the study context including the reason for undertaking the study, age-group and study design, weren't entirely clear.

With regards to the criteria for publication:

1. The study appears to present the results of original research.

2. Results reported do not appear to have been published elsewhere.

3. Experiments, statistics, and other analyses are performed to a high technical standard and are described in sufficient detail, excepting the study question and context could be refined.

--

Thank you very much for your feedback. We complemented and specified the study background to further explain why we focused on the age group 80+, found there was a need to perform a qualitative explorative study among physicians and dentists, and also on the reasons to choose the specific design of a qualitative survey. Details including additional references were added in the introduction (p. 4, lines 52-53, 63-69, p. 5 lines 83-87, 98-99, p. 6 lines 100-105) and the methods section (p. 7, lines 137-140).

--

2.

4. Conclusions are presented in an appropriate fashion and are supported by the data.

5. The article is presented in an intelligible fashion and is written in standard English.

6. The research appears to meet all applicable standards for the ethics of experimentation and research integrity.

7. The article adheres to appropriate reporting guidelines and community standards for data availability. I think the structure is clear and likely conforms to the main requirements. For studies that clearly meet a reporting format, I always suggest that the authors double-check conformance with that checklist, in this case it would be CROSS

(https://www.equator-network.org/reporting-guidelines/a-consensus-based-checklist-for-reporting-of-survey-studiescross/).

--

Thank you for suggesting CROSS as reporting guideline. We were uncertain about referrals to reporting guidelines since there is none specifically for qualitative surveys. Because generally, this is a qualitative study, we primarily relied on COREQ. However, to ensure proper reporting of the elements of a survey design, we now also consulted CROSS and incorporated all information requirements as far as they were applicable to a qualitative survey. We added this explanation in the methods section (p. 6, lines 117-120). Moreover, we added details regarding response rates and repeated participation in the strengths and limitations section after checking CROSS (p. 33, lines 591-596). 

--

Comments of reviewer 1

1. Manuscript by Herrler et al is well written. It’s an important survey which explains what outpatient healthcare systems require to provide proper care to patients aged 80 and over. I have few concerns before manuscript accepted for publication.

--

Thank you very much for your appreciation. 

--

2. Abstract: Line 28, how many ambulatory healthcare sectors were included in the survey. It would reflect the global scenario of the survey being held.

--

Thank you for this suggestion. We added the information that all medical specialties apart from child and adolescent healthcare we eligible (p. 2, lines 29-30). Moreover, we complemented information on the finally included specialties (p. 2, lines 34-36). 

--

3. Line 34; Author mentioned, ‘Caring for patients aged 80 and over was perceived as challenging because of its complexity…’ Are authors talking about the complexity of disease? Please elaborate.

--

Thank you for this comment. We specified what is meant by complexity on page 2, lines 37-38.

--

4. This survey was carried out in year 2021-2022. COVID situation must influence overall outpatient resources and care facility. Author could discuss this aspect in discussion section.

--

Thank you very much for this important remark. We complemented the discussion with an overview on how the pandemic affected ambulatory healthcare in Germany in general, and what this might mean for the interpretation of our results (p. 30-32, lines 540-574).

--

5. Format of references should be consistent. Check references 28, 31, 45

--

Thank you for this comment, we checked the references accordingly.

--

6. All images are blurred. High quality images are required

--

Thank you for this comment. This seems to be a problem of implementation while the complete submission PDF is built. When clicking on “access/download figure”, the figures appeared as intended. We will carefully check the appearance of the figures during the resubmission process.

--

---

## [Editor Report · Decision Letter 1]

28 Jul 2022

Providing ambulatory healthcare for people aged 80 and over: views and perspectives of physicians and dentists from a qualitative survey

PONE-D-22-11053R1

Dear Dr. Herrler,

We’re pleased to inform you that your manuscript has been judged scientifically suitable for publication and will be formally accepted for publication once it meets all outstanding technical requirements.

Kind regards,

Dylan A Mordaunt, MD, MPH, FRACP

Academic Editor

PLOS ONE

Additional Editor Comments (optional):

Thank you for your resubmission. This now meets the criteria for publication.
---

## [Editor Report · Acceptance letter]

5 Aug 2022

PONE-D-22-11053R1 

Providing ambulatory healthcare for people aged 80 and over: views and perspectives of physicians and dentists from a qualitative survey 

Dear Dr. Herrler:

I'm pleased to inform you that your manuscript has been deemed suitable for publication in PLOS ONE. Congratulations! Your manuscript is now with our production department. 

Kind regards, 

on behalf of

Associate Professor Dylan A Mordaunt 

Academic Editor

PLOS ONE